# Monthly variance in UK renal transplantation activity: a national retrospective cohort study

Marcus Lowe,[1] Robert Maidstone,[2,3] Kay Poulton,[1] Judith Worthington,[1] Hannah J Durrington,[2,4] David W Ray,[3,5] David van Dellen,[2,6] Argiris Asderakis,[7] John Blaikley,[2,4] Titus Augustine[2,6]

For numbered affiliations see end of article.

**Correspondence to**
Dr John Blaikley;
john.blaikley@manchester.ac.uk

Prof. Titus Augustine;
titus.augustine@mft.nhs.uk

## ABSTRACT

**Objective** To identify whether renal transplant activity varies in a reproducible manner across the year.

**Design** Retrospective cohort study using NHS Blood and Transplant data.

**Setting** All renal transplant centres in the UK.

**Participants** A total of 24 270 patients who underwent renal transplantation between 2005 and 2014.

**Primary outcome** Monthly transplant activity was analysed to see if transplant activity showed variation during the year.

**Secondary outcome** The number of organs rejected due to healthcare capacity was analysed to see if this affected transplantation rates.

**Results** Analysis of national transplant data revealed a reproducible yearly variance in transplant activity. This activity increased in late autumn and early winter (p=0.05) and could be attributed to increased rates of living (October and November) and deceased organ donation (November and December). An increase in deceased donation was attributed to a rise in donors following cerebrovascular accidents and hypoxic brain injury. Other causes of death (infections and road traffic accidents) were more seasonal in nature peaking in the winter or summer, respectively. Only 1.4% of transplants to intended recipients were redirected due to a lack of healthcare capacity, suggesting that capacity pressures in the National Health Service did not significantly affect transplant activity.

**Conclusion** UK renal transplant activity peaks in late autumn/winter in contrast to other countries. Currently, healthcare capacity, though under strain, does not affect transplant activity; however, this may change if transplantation activity increases in line with national strategies as the spike in transplant activity coincides with peak activity in the national healthcare system.

## INTRODUCTION

When organ donors and families of potential donors consent to donation, they make a very valuable gift which is life-changing for the recipient. Despite recent improvements in the UK[1] and other countries regarding access to transplantation, demand for renal transplantation still exceeds the number of available donors.[2] This results in increased

### Strengths and limitations of this study

► The study was a 10-year retrospective study involving all the registered renal transplant recipients in the UK over this time period.
► The national transplant database is filled using data submitted by each transplant centre; therefore, the data has not been independently corroborated.
► A minority of patients will have refused to be enrolled in the national database potentially affecting our findings.
► The database is used to audit transplant provision and outcomes in the UK; therefore, it was not set up specifically for this research project.

patient mortality and morbidity.[3] Since many potential donors cannot be used for various reasons, it is vital that these reasons are minimised so that every potential organ for transplant is used for the primary intended recipient. One of the reasons transplantation does not proceed is due to a lack of clinical capacity. Therefore, healthcare planning plays a key role in ensuring that sufficient capacity exists so that all transplants are used for the primary intended recipients. This could be potentially difficult as the National Health Service (NHS) often works near or at maximum capacity,[4] especially in winter months.

Renal transplantation uses both living and deceased donors. Human mortality rates are known to oscillate in a seasonal manner for some diseases.[5] This can be attributed to the effects of endogenous seasonal rhythms and climatic factors on human performance and activity patterns. In the UK, seasonal variations are commonly observed for infectious diseases, such as influenza,[6] cerebrovascular disease[7–12] and myocardial infarctions.[13] Several studies have recently shown how understanding these oscillations are crucial for planning and delivering optimum

healthcare delivery.[14] For example, the USA covers many different climate zones, but depite this cerebrovascular accidents in the USA peak in the winter[15] in a similar pattern to the UK.[7–12] Surprisingly, transplant activity in the USA is lowest in the winter months for both renal[16] and heart transplantation, despite these deaths being a leading cause of organ donation.[17] A similar pattern is also seen in Italy,[18] suggesting that this seasonal pattern is conserved in other countries.

To the best of our knowledge, seasonal fluctuations in transplant activity have not been investigated in the UK. We, therefore, examined the national cohort over a 10-year period to establish both seasonal fluctuations and whether healthcare capacity influences activity.

## METHODS

All UK renal adult transplants, performed between 2005 and 2014 to recipients over 18 years old, were included in the study. Data were provided by NHSBT (NHS Blood and Transplant) who maintain a comprehensive national database on the 24 adult kidney transplant centres in the UK. This data was combined with data collected on donor activity, provided by the National Organ Retrieval Service. Deceased donation is categorised into donation after brain death and donation after circulatory death. All donor deaths are further classified according to the cause of mortality. 'Cerebrovascular event' was defined as death

due to intracranial haemorrhage, intracranial thrombus or unclassified intracranial event. 'Infection' was defined as death due to meningitis, septicaemia, pneumonia or unclassified infection. Deceased donor transplants form about two-thirds of all UK kidney transplant activity. The other one-third of kidney transplants are transplants from living donors. Living donor transplants are logistically different as they are planned and scheduled elective cases depending on several factors which can usually be controlled. Deceased donor transplants, on the other hand, are unplanned occurring when organs become available after the death of a suitable donor. Recipients are then allocated organs according to agreed national allocation criteria.

Data were analysed according to the month and year the transplant occurred. Donation details were analysed in a similar manner. The observed data were compared with the expected transplant activity calculated by measuring the total number of transplants divided by the number of days in the same year.

Data were also analysed according to organs offered to different centres for named patients and declined, for various reasons by the centre, including capacity issues. These issues for the purposes of this paper were subclassified into different categorical values (centre already transplanting, no beds, no staff, no theatre and no time).

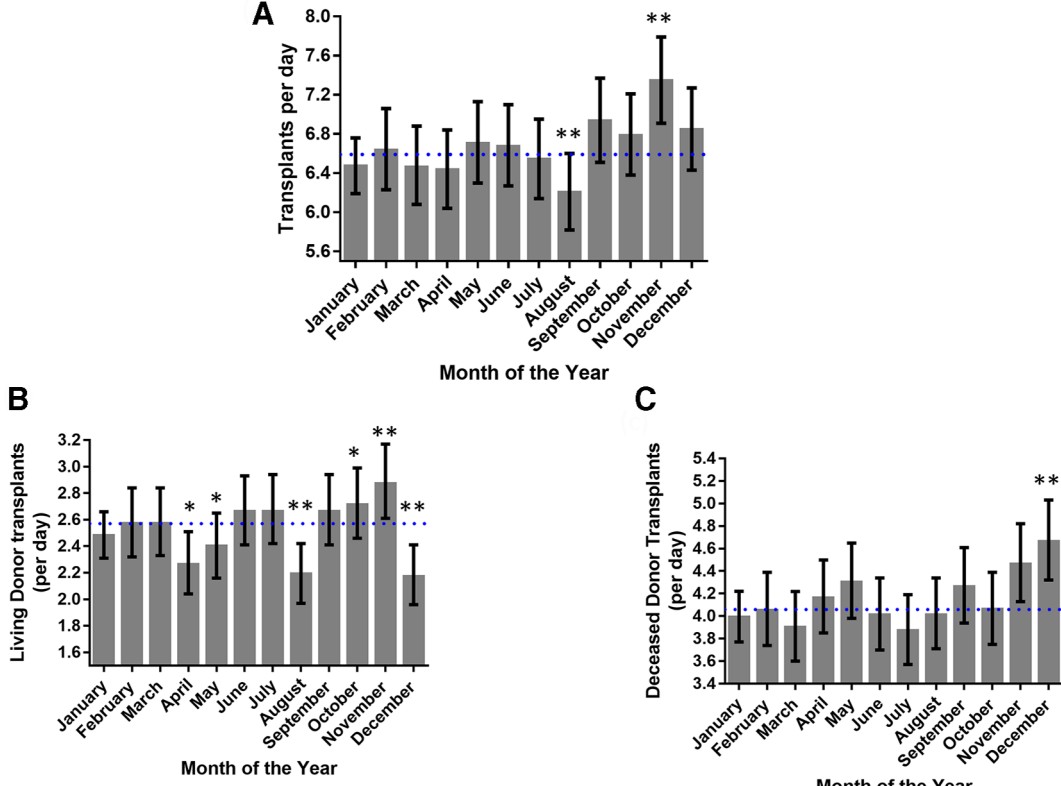

**Figure 1** Changes in transplant activity during the year. (A) The number of UK renal transplants per month varied significantly during the year (p<0.01) peaking in November. (B) Renal transplants arising from living donors varied significantly throughout the year peaking in October and November. (C) Transplant activity using deceased donors varied significantly over the year peaking in December. (*P<0.05, **p<0.01 Pearson residual, dotted line shows average activity).

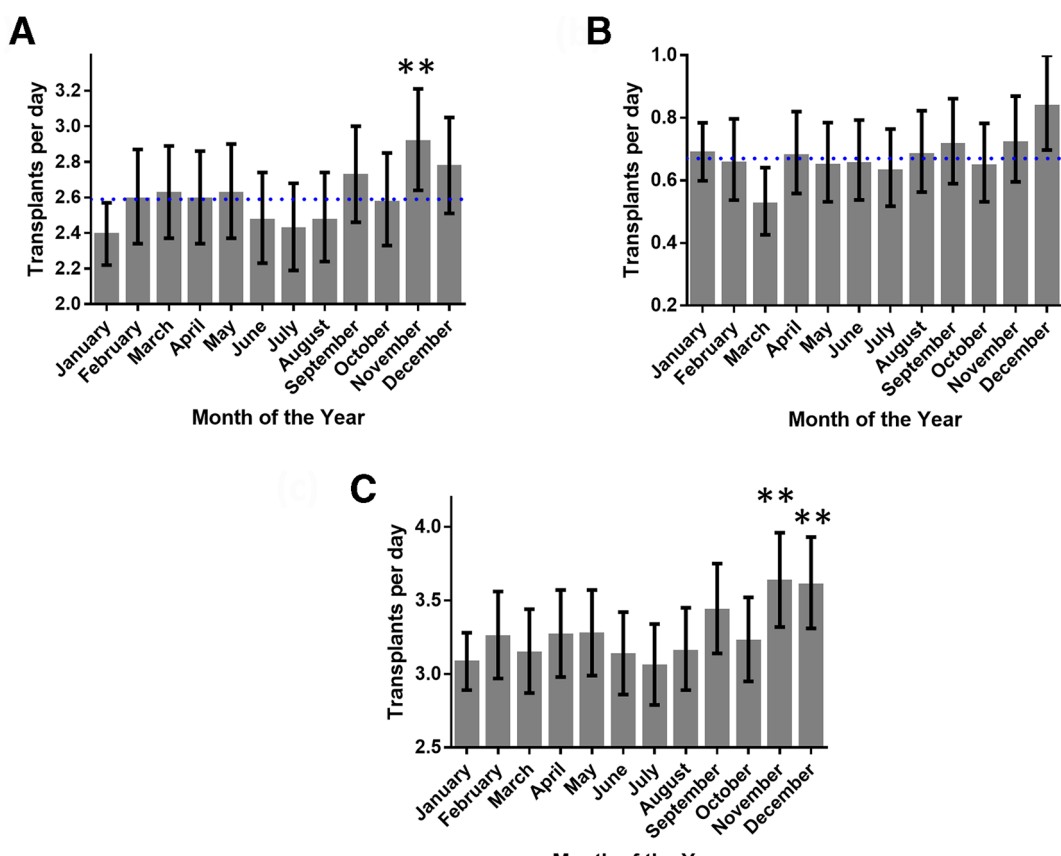

**Figure 2** Changes in transplant activity using donors after cerebrovascular events or hypoxic brain injury. (A) Transplant activity using donors following cerebrovascular events peaks in November. (B) Transplant activity using donors with hypoxic brain damage also increased in activity at the end of the year (November and December). (C) By combining both these causes, the graph mirrors the fluctuation seen in transplant activity for all deceased donors (shown in figure 1C). (*P<0.05, **p<0.01 Pearson residual, dotted line shows average activity).

### Statistical methods

Generalised linear models (GLMs), with a Poisson link function, were used to model the effect that month of the year had on our data. Offset variables were used to account for the slight differences in the number of days per month. To account for the unknown correlation structure in our data, generalised estimating equations (GEEs) per year were used to estimate the GLM parameters.[19]

The $\chi^2$ goodness-of-fit tests assessed whether the observed number of transplants per month differed from the expected value if there was no seasonal variability. When this test was significant (ie, the expected values were significantly different from the observed values), Pearson residuals were used to identify the particular months which caused this mismatch. Residuals that are greater than two in magnitude suggest some degree of lack of fit.[20]

Sine and cosine curves were fitted to the transplant data to investigate whether repeating patterns occurred across the months. All analyses were performed using SPSS V.22 (IBM corp.).

### Patient and public involvement

Patients and the public were not involved in the design or analysis of data for this study.

### Ethics

Data were collected by NHSBT and analysed after obtaining approval from them in accordance with their policies. Since the data were fully anonymised, no ethical permission was sought.

### RESULTS

During the 10-year study period, 24 270 adult kidney transplants were performed in the UK. A total of 15 094 (62%) were from deceased donors and 9166 (38%) were from living donors.

### Kidney transplant activity changes in a consistent manner throughout the year

Kidney transplant activity varied by 17.1% within a year (p<0.01, $\chi^2$, figure 1A). Transplant activity was highest in November (11% increase from mean, p<0.01) and lowest in August (6% decrease from mean, p<0.01). Investigating seasonal variation, transplant activity increased in the autumn compared with spring and summer (p<0.05, one-way analysis of variance); these patterns were consistent every year during the 10-year study period (p<0.01, GEE).

Kidney transplantation uses both living and deceased donors. Our data set was, therefore, examined to see

**Table 1** Seasonal variation in kidney transplants using organs from deceased donors

| Cause of death in the donor | n | P value for whether donation varied across the year |
|---|---|---|
| Intracranial/cerebrovascular causes | 9452 | <0.01 |
| Hypoxic brain damage | 2448 | <0.01 |
| Trauma | 1532 | <0.01 |
| Infective causes | 367 | <0.01 |
| Cardiovascular | 237 | 0.13 |
| Respiratory/pulmonary causes | 192 | 0.17 |
| Brain tumours | 180 | 0.14 |
| Poisoning/drug overdose | 55 | 0.03 |
| Organ failure (various causes) | 53 | <0.01 |
| Other/unknown | 578 | 0.01 |

All deceased donations over a 10-year period were split into groups according to aetiology defined by NHS Blood and Transplant. The incidence for the majority (7/10) of causes varied significantly during the year.

**Table 3** Seasonal variation in kidney transplants using organs from donors after circulatory death

| Cause of death in the donor | n | P value for whether donation varied across the year |
|---|---|---|
| Intracranial/cerebrovascular causes | 2467 | 0.60 |
| Hypoxic brain damage | 1216 | 0.16 |
| Trauma | 562 | <0.01 |
| Infective causes | 107 | 0.05 |
| Cardiovascular | 166 | 0.88 |
| Respiratory/pulmonary causes | 185 | 0.25 |
| Brain tumours | 31 | 0.07 |
| Poisoning/drug overdose | 29 | <0.01 |
| Other/unknown | 228 | <0.01 |

All donations after circulatory death over a 10-year period were split into groups according to aetiology defined by NHS Blood and Transplant. The incidence for a minority (3/9) of causes varied significantly during the year.

if one of these types was responsible for the variance in activity. Transplant activity using organs from living donors (figure 1B) significantly increased in October and November and decreased in December, April, May and August. Transplant activity using organs from deceased donors (figure 1C) increased in December and a similar trend was observed in November. Therefore, the increase in activity for November, which has the highest activity, is due to an increase in transplants using kidneys from both living and deceased organ donors. This contrasts with August, which had the lowest activity, and is solely explained by a fall in living donor activity.

### Incidence for brain injury donors fluctuates over the year
Donation from cerebrovascular and hypoxic brain injury accounted for 78.8% of all deceased donor transplants; therefore, we hypothesised that the spike in deceased donor transplants is due to an increase in donors from these categories during November and December. Cerebrovascular deaths, as a cause of donor death, were significantly increased in November (12% increase from the mean, p<0.01) (figure 2A). Donations from donors who had experienced a hypoxic brain injury tended to increase during this period with the highest spike being in December (25% increase from the mean, p=0.15) (figure 2B). When these results are combined, they mirror the fluctuations seen for all deceased organ transplant activity (figure 2C) with significant elevations in transplant activity during November and December (p<0.01).

### Seasonal variance in the incidence of donors after infection
We also noted that other causes of donor death significantly varied over the year (table 1). Interestingly, this variance was mainly observed in donations after brain death (table 2) rather than circulatory death (table 3). Donations after road traffic accidents (figure 3A) were higher during late spring, summer and early autumn (April–September) compared with other times of the year. Donations from donors dying from infective causes peaked in the winter and declined in the summer (figure 3B); this relationship could be explained by a cosine curve (figure 3B, p<0.05) suggesting a potential underlying seasonal oscillation.

**Table 2** Seasonal variation in kidney transplants using organs from donors after brain death

| Cause of death in the donor | n | P value for whether donation varied across the year |
|---|---|---|
| Intracranial/cerebrovascular causes | 6985 | <0.01 |
| Hypoxic brain damage | 1232 | <0.01 |
| Trauma | 970 | <0.01 |
| Infective causes | 260 | <0.01 |
| Cardiovascular | 71 | <0.01 |
| Respiratory/pulmonary causes | 7 | 0.18 |
| Brain tumours | 149 | 0.57 |
| Poisoning/drug overdose | 26 | <0.01 |
| Other/unknown | 350 | 0.21 |

All donations after brain death over a 10-year period were split into groups according to aetiology defined by NHS Blood and Transplant. The incidence for the majority (6/9) of causes varied significantly during the year.

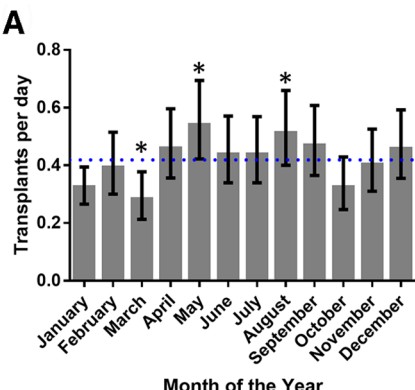
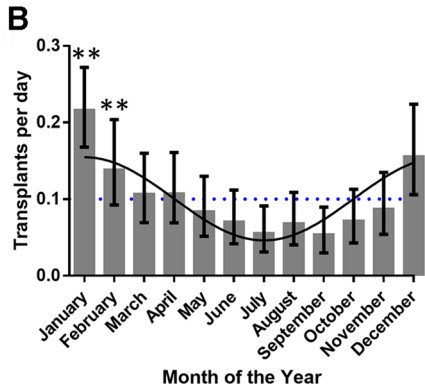

**Figure 3** Seasonal variation in transplant activity arising after infection or trauma. (A) Transplant activity using donors following traumatic incidents increased in the summer (B) as opposed to transplant activity using donors following infection, which oscillated in a sinusoidal manner peaking in the winter. (*P<0.05, **p<0.01 Pearson residual, best fit line for infection is a cosine (p<0.05), dotted line shows average activity).

## The effect of healthcare capacity on transplant activity

The UK healthcare system routinely works at high capacity,[21] potentially resulting in reduced transplantation rates due to bed and staffing shortages. We, therefore, examined whether the reproducible monthly changes in transplant activity could be attributed to a lack of capacity in the healthcare system. Although a large number 28 789 of deceased donor offers were declined for transplantation over the study period only a small fraction of these were due to lack of capacity (n=480, mean 1.22% (±0.22% SD)).

## DISCUSSION

The NHS in the UK is coping with increased pressures, especially in the winter.[21] This study reveals that UK kidney transplant activity peaks in November and December, at the beginning of the winter, in contrast with a number of European[18 22] and North American centres.[16 17] This unexpected finding is due to variations in living and deceased donation. Although detailed causal analysis was beyond the scope of the study, the winter increases in deceased donation could be attributed to increases in both cerebrovascular and hypoxic brain events. This is consistent with findings from previous studies studying the incidence of these events both in the UK[9] and other northern hemisphere countries.[11 12]

The winter surge in transplant activity has important implications for the UK health system (NHS). During the UK winter, there is also a surge in emergency admissions to hospitals, placing the system under significant strain, sometimes resulting in cancellation of elective and semi-elective operations for up to several months.[23] It was, therefore, reassuring that we found at the national level no solid evidence of transplant surgery using deceased donors being cancelled during the winter months. Despite this, however, it is important that individual transplant departments plan for this predictable and reproducible surge in transplant activity making sure that their own activity is not affected. We are confident that this surge

is likely to continue into the future as two different statistical tests ($\chi^2$ and GEE) produced similar results. This is especially important as UK transplant activity is likely to increase due to recent changes in legislation.

Certain limitations should be noted when interpreting the results of this study. The donor rates reported in the paper are from used donations, and therefore can be influenced by changes in donor conversion rates; the proportion of donor offers used in a transplant operation. This is unlikely, however, as these changes would have to be consistently occurring in the same way each year for 10 years; furthermore, the peak of deceased donation activity coincided with the epidemiological peaks for the underlying diseases.[7–12] Since this was a retrospective study, there is always the potential for inherent bias, but since this is relatively a large data set, it should minimise this effect. The data from each centre could not be independently verified, due to anonymisation in the data set. Finally, the effect of healthcare capacity on living donations was not investigated, but this would be an important area to investigate in future research.

Our study clearly shows that UK renal transplant activity increases in the winter in contrast to previous studies investigating seasonal transplant activity in other countries. This could have implications for the UK health system since winter is when the British health system is placed under maximal strain. This seasonal variation should, therefore, be considered for any future planning especially with the potential impact of the opt-out legislation and strategies to increase organ donation and transplantation.

**Author affiliations**
[1]Transplantation Laboratory, Manchester University NHS Foundation Trust, Manchester, UK
[2]Faculty of Biology, Medicine and Health, University of Manchester, Manchester, UK
[3]Oxford Centre for Diabetes, Endocrinology and Metabolism, University of Oxford, Oxford, UK
[4]Department of Respiratory Medicine, Manchester University NHS Foundation Trust, Manchester, UK
[5]NIHR Oxford Biomedical Research Centre, John Radcliffe Hospital, Oxford, UK

⁶Transplant and Endocrine Surgery, Manchester University NHS Foundation Trust, Manchester, UK
⁷Cardiff Transplant Unit, University Hospital of Wales, Cardiff, UK

**Acknowledgements** The authors would like to acknowledge Phil Foden for his contribution to the statistical analysis of the results, and NHSBT and UK renal transplant centres for collecting and providing the data.

**Contributors** TA and JB conceived the study. TA, JB and ML obtained and analysed the data. RM did the statistical analysis. KP, JW, HJD, DvD, DWR and AA substantially contributed to the interpretation of the data. TA, JB and AA drafted the manuscript, with the other authors who revised it critically.

**Funding** JB holds an MRC Clinician Scientist Award (MR/L006499/1). DWR holds a Wellcome Trust Investigator Award (107851/Z/15/Z) and a Medical Research Council grant (MR/P023576/1). HJD is supported by an Asthma UK Senior Clinical Academic Development Award (AUK-SCAD-2013-229). JB and HJD are partially supported by the Manchester NIHR Biomedical Research Centre.

**Competing interests** None declared.

**Patient consent for publication** Not required.

**Provenance and peer review** Not commissioned; externally peer reviewed.

**Data availability statement** Data are available on reasonable request, but may be subject to approval from NHSBT. The statistical code has been uploaded onto a data repository (doi: https://dx.doi.org/10.17632/48nxwvcfnh.1).

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
