## [Reviewer comments · BMJ Open]

ARTICLE DETAILS

TITLE (PROVISIONAL)	Monthly variance in United Kingdom renal transplantation activity: a national retrospective cohort study
AUTHORS	Lowe, Marcus; Maidstone, Robert; Poulton, Kay; Worthington, Judith; Durrington, Hannah J.; Ray, David; van Dellen, David; Asderakis, Argiris; Blaikley, John; Augustine, Titus

VERSION 1 – REVIEW

REVIEWER	Axel Rahmel. MD Deutsche Stiftung Organtransplantation (German Organ Procurement Organisation) - Medical Director - Frankfurt a.M. Germany
REVIEW RETURNED	11-Feb-2019

GENERAL COMMENTS	The authors intend to address two different questions: 1.) Does organ donation and renal transplantation show seasonal variance in the UK2.) Are (kidney) transplant rates affected by healthcare capacity Ad 1.) As far as can be judged by the Methods and Results section, donor rates for the different causes of death were based on the number of utilized donors in the different months. It would be important to look at the number of reported donors for each of these categories, because differences in overall acceptance rates in the different months could influence this finding. Possible confounding factors like donor age distribution over the different months (reported donors, utilized donors) should be looked at to see, whether this affects the conclusions. Ad 2.) The analysis of the influence of healthcare capacity is dependent on the reliability of the data provided by the transplant centers regarding the number of turndowns based on lack of capacity. If a center reports a high rate of turndowns for this reason, patients, referring doctors and authorities might lose trust in this center. Therefore, it seems likely that underreporting of this reason for turning down an organ offer took place. It could be possible, that the kidneys of a donor were officially not accepted for “donor quality” reasons but in fact, “transplant capacity” was not available. Are there any data on the number of organs that were turned down due to poor organ quality by one center and were then transplanted in another center?
--

	These points should be addressed to better understand, whether the conclusions of this retrospective study are justified. Additional specific points: Page 6/23, line 18: The authors claim that there was an increase in patient mortality and morbidity in recent years. A reference for this statement should be provided and it should be clarified whether this statement refers to patients on the waiting list or patients with ESRD. Page 8/23, line 30: It should read, "Deceased donor transplants form about two thirds of all UK kidney transplant activity..." The figures are small and difficult to read, comparison between transplant and donation rates are difficult to make. The authors should consider adding a figure that shows the relative contribution of the different causes of donor death to the overall number of donors in the different months.
--	--

REVIEWER	Khaled Tafran University of Malaya, Malaysia
REVIEW RETURNED	16-Feb-2019

GENERAL COMMENTS	@Overall # This is an interesting study investigating the seasonal variations in renal transplant activities in the UK between 2004 and 2014. # The study is well-conducted and written and the results are of some importance to policy makers. However, there are some minor corrections that may make the study better, in my opinion. @ Results and data analysis # The statistical method used in this study is adequate and convincing; however, the authors did not report the results in detail; i.e. the GEEs and GLM models. The readers may need additional information to better understand the results. Perhaps, the authors did not report the results in detail due to word count limits, so I suggest adding more details about the results as supplementary materials. @ Limitations # The study uses aggregate data for all renal transplant activities in the UK; however, the results of this study (seasonal variations) may not necessarily be true for some transplant centers. Therefore, at the micro-level (transplant center), the results of this study would be useful, but further center-specific analysis is also important. The authors need to address this issue as a limitation of this study. @ Writing and reporting # The article is well written, however, there are some missed commas in the introduction, in the following sentences: "Despite recent improvements in the United Kingdom (UK) and other countries regarding access to transplantation demand for renal transplantation exceeds the number of available donors leading to increased patient mortality and morbidity."
---

	"Since many potential donors cannot be utilised for various reasons it is vital that these are minimised so that every potential organ for transplant is utilised for the primary intended recipient." # The comma in the following sentence should be deleted: "The other one third of kidney transplants, are transplant from living donors."
--	---

REVIEWER	Vaishaly Bharambe Dr D Y Patil Medical College, Hospital and Research center, Dr D Y Patil Vidyapeeth, Pune, India
REVIEW RETURNED	25-Mar-2019

GENERAL COMMENTS	With full respect to the eminent authors of this manuscript I would like to state that, I am unable to understand the need for this study which is basically an easily drawn statistical inference which can be completed in maybe 15-20 lines. It does not justify its being labeled as a research article. However the finding could be significant and could be made note of in some other form of publication.
--

REVIEWER	Wai Lim University of Western Australia and Sir Charles Gairdner Hospital, Perth, Australia
REVIEW RETURNED	15-Apr-2019

GENERAL COMMENTS	Population cohort examining the pattern and variation in UK adult kidney transplant activity between 2005-2014. The authors systematically examined the change in LD and DD donation rates, along with variation in DD causes of death and related transplantation activity. Largely a well-designed/well-written paper with appropriate analyses. However, there are multiple factors that are likely to result in the variation in donation/transplantation activity over the year, some of which are not captured using national databases. I would like the authors to address the issues below:  1) LD rate or variation – due to intrinsic donor/patient factors or factors within the healthcare system? Surgery availability (school holidays and hence surgical availability). Do the authors have donor relationship to recipients? 2) Did (or can) the authors account for centre effect or era variations (patterns may have changed over time especially changes in donor characteristics and acceptance criteria) 3) Can the authors comment further on the variation in infection deaths (DD) – cause of death recorded? 4) What about DCD donors/transplant activity? 5) Do the authors have data to allow differentiation into lack of capacity (to undertake transplantation) vs discards? Do the authors have data on “missed donors” that may give a better insight into the potential seasonal change in transplantation activity?
--

VERSION 1 – AUTHOR RESPONSE

Reviewer: 1 01204 492059

Reviewer Name: Axel Rahmel. MD

Institution and Country: Deutsche Stiftung Organtransplantation, (German Organ Procurement Organisation)- Medical Director - Frankfurt a.M.Germany Please state any competing interests or state 'None declared': None declared

Please leave your comments for the authors below The authors intend to address two different questions:

- 1.) Does organ donation and renal transplantation show seasonal variance in the UK
- 2.) Are (kidney) transplant rates affected by healthcare capacity

Ad 1.)

As far as can be judged by the Methods and Results section, donor rates for the different causes of death were based on the number of utilized donors in the different months. It would be important to look at the number of reported donors for each of these categories, because differences in overall acceptance rates in the different months could influence this finding. Possible confounding factors like donor age distribution over the different months (reported donors, utilized donors) should be looked at to see, whether this affects the conclusions.

We agree that the donor rates reported in the paper are from actual transplants that have been carried out and therefore total donor offer rates were not examined. We also agree that changes in donor acceptance rates could be a potential explanation for our findings if we only studied one year. Our cohort however was over ten years with the pattern staying the same for each individual year during this period making it very unlikely that donor acceptance rates skewed our findings. Furthermore, other epidemiological studies have confirmed the winter spike in incidence for the main cause of donors i.e. hypoxic brain injury or intracranial events as mentioned in the introduction

Ad 2.)

The analysis of the influence of healthcare capacity is dependent on the reliability of the data provided by the transplant centers regarding the number of turndowns based on lack of capacity. If a center reports a high rate of turndowns for this reason, patients, referring doctors and authorities might lose trust in this center. Therefore, it seems likely that underreporting of this reason for turning down an organ offer took place. It could be possible, that the kidneys of a donor were officially not accepted for "donor quality" reasons but in fact, "transplant capacity" was not available. Are there any data on the number of organs that were turned down due to poor organ quality by one center and were then transplanted in another center?

We do agree that a potential limitation of any study, not only ours, is the reliability of data. To ensure that our data was robust as possible it was collected by a nationally recognised body, independent of the investigators, furthermore where a transplant centre declines a donor to a named recipient the data is audited by NHSBT. We now make this clearer by changing the limitations of the study to "The national transplant database is filled using data submitted by each transplant center; therefore the data has not been independently corroborated."

Even if we had the data on organs that have been turned down by one centre but used in another centre it would be difficult to separate out the nuances associated with the organ declines clearly establishing that the reason was due to lack of capacity. This is because each centre accepts different quality of organs depending on local expertise therefore an organ that is deemed unsuitable at one centre may be suitable at another.

Ultimately transplant teams desire what is best for the donor and recipient i.e. a successful transplant and robust reporting of outcomes is an important part of this process.

These points should be addressed to better understand, whether the conclusions of this retrospective study are justified.

Additional specific points:

Page 6/23, line 18:

The authors claim that there was an increase in patient mortality and morbidity in recent years. A reference for this statement should be provided and it should be clarified whether this statement refers to patients on the waiting list or patients with ESRD.

This statement was intended to say that since demand is higher than supply, not everyone will get a kidney transplant who needs one. This therefore results in increased patient morbidity and mortality. This has been updated with the correct references in the manuscript.

Page 8/23, line 30:

It should read, "Deceased donor transplants form about two thirds of all UK kidney transplant activity..."

We agree and this has now been corrected

The figures are small and difficult to read, comparison between transplant and donation rates are difficult to make. The authors should consider adding a figure that shows the relative contribution of the different causes of donor death to the overall number of donors in the different months.

We did consider doing this however due to concerns about presenting the same data twice, we feel it is best to present the data in its current format since this allows the reader to clearly see temporal changes in activity per cause of donor death. However the requested graph is presented below for the reviewers information:

Reviewer: 2

Reviewer Name: Khaled Tafran

Institution and Country: University of Malaya, Malaysia Please state any competing interests or state 'None declared': None declared

Please leave your comments for the authors below @Overall # This is an interesting study investigating the seasonal variations in renal transplant activities in the UK between 2004 and 2014. # The study is well-conducted and written and the results are of some importance to policy makers. However, there are some minor corrections that may make the study better, in my opinion.

@ Results and data analysis

The statistical method used in this study is adequate and convincing; however, the authors did not report the results in detail; i.e. the GEEs and GLM models. The readers may need additional information to better understand the results. Perhaps, the authors did not reported the results in detail due to word count limits, so I suggest adding more details about the results as supplementary materials.

In order to achieve this we have now uploaded the statistical code into a data repository. If readers would like to know more information, then of course they can contact either corresponding author who will provide the information.

@ Limitations

The study uses aggregate data for all renal transplant activities in the UK; however, the results of this study (seasonal variations) may not necessarily be true for some transplant centers. Therefore, at the micro-level (transplant center), the results of this study would be useful, but further center-specific analysis is also important. The authors need to address this issue as a limitation of this study.

This has now been added to the manuscript as below:

It was therefore reassuring that we found at the national level no solid evidence of transplant surgery utilising deceased donors being cancelled during the winter months. Despite this however it is important that transplant departments plan for this predictable and reproducible surge in transplant activity making sure their own activity is not affected.

@ Writing and reporting

The article is well written, however, there are some missed commas in the introduction, in the following sentences:

"Despite recent improvements in the United Kingdom (UK) and other countries regarding access to transplantation demand for renal transplantation exceeds the number of available donors leading to increased patient mortality and morbidity."

"Since many potential donors cannot be utilised for various reasons it is vital that these are minimised so that every potential organ for transplant is utilised for the primary intended recipient."

The comma in the following sentence should be deleted:

"The other one third of kidney transplants, are transplant from living donors."

We agree and thank the reviewer for pointing this out. It has now been corrected in the manuscript.

Reviewer: 3

Reviewer Name: Vaishaly Bharambe

Institution and Country: Dr D Y Patil Medical College, Hospital and Research center, Dr D Y Patil Vidyapeeth, Pune, India Please state any competing interests or state 'None declared': None Declared

Please leave your comments for the authors below With full respect to the eminent authors of this manuscript I would like to state that, I am unable to understand the need for this study which is basically an easily drawn statistical inference which can be completed in maybe 15-20 lines. It does not justify its being labeled as a research article. However the finding could be significant and could be made note of in some other form of publication.

We appreciate reviewer's 3 interest in the manuscript and recognition of the significance of our work. Since other reviewers have asked for more information, we have added extra information to meet differing viewpoints. Hopefully for readers seeking a summary of our key findings the abstract should suffice.

Reviewer: 4

Reviewer Name: Wai Lim

Institution and Country: University of Western Australia and Sir Charles Gairdner Hospital, Perth, Australia Please state any competing interests or state 'None declared': None

Please leave your comments for the authors below Population cohort examining the pattern and variation in UK adult kidney transplant activity between 2005-2014. The authors systematically examined the change in LD and DD donation rates, along with variation in DD causes of death and related transplantation activity. Largely a well-designed/well-written paper with appropriate analyses. However, there are multiple factors that are likely to result in the variation in donation/transplantation activity over the year, some of which are not captured using national databases.

I would like the authors to address the issues below:

- 1) LD rate or variation – due to intrinsic donor/patient factors or factors within the healthcare system? Surgery availability (school holidays and hence surgical availability). Do the authors have donor relationship to recipients?

Unfortunately, we do not have access to this data and this is one of the reasons why we state in the discussion that a detailed causal analysis is beyond the scope of this manuscript. Since the paper is focused on establishing whether any significant variation exists rather than establishing a causal relationship we do not feel the lack of this analysis subtracts from the message of the overall paper.

- 2) Did (or can) the authors account for centre effect or era variations (patterns may have changed over time especially changes in donor characteristics and acceptance criteria)

We agree that these are important points and have tried to address them below:

There are centre variations in the acceptance of higher donor risk index kidneys and the ration of DBD and DCD kidneys accepted. These are demonstrated very clearly in the annual centre specific reports produced by NHSBT. Despite this centre variation it cannot explain the national variation reported in the manuscript since all rejected kidneys are offered nationally, allowing other centres to accept them for transplantation.

Era variations were accounted for by creating ten separate GLMs, one for each year under investigation. No significant variations in patterns between these models were found and we have made this clearer in the methods section:

“To account for the unknown correlation structure in our data, generalised estimating equations (GEEs) per year were used to estimate the GLM parameters”

- 3) Can the authors comment further on the variation in infection deaths (DD) – cause of death recorded?

Infectious deaths were due to meningitis (n=280), pneumonia (n=61), septicaemia (n=17) and were unspecified in nine cases. The small numbers preclude any robust seasonal variation for the subtypes.

- 4) What about DCD donors/transplant activity?

We thank the reviewer for their suggestion and have performed the suggested analysis. This shows that seasonal variation mainly affects donation after brain death rather than donation after circulatory arrest. The results are presented in two additional tables (table 2 and 3) and the following sentence has been added to the text in the section titled seasonal variance.

“This was mainly seen in donation after brain death (table 2) rather than circulatory death (table 3).”

- 5) Do the authors have data to allow differentiation into lack of capacity (to undertake transplantation) vs discards? Do the authors have data on “missed donors” that may give a better insight into the potential seasonal change in transplantation activity?

35 transplants were declined based on the fact the centre was already retrieving or transplanting. This is a small fraction of the 480 transplants declined for capacity issues. Unfortunately, we do not have further granular data regarding lack of capacity.

With regard to missed donors, The UK national Potential Donor Audit (PDA) commenced in 2003 as part of a series of measures to improve organ donation and to ensure no potential donors were “missed”. The principal aim of this audit was to determine the potential number of solid organ donors in the UK and provide information about the hospital practices surrounding donation. The latest annual report for the financial year, 1st April 2017 to 31st March 2018, shows that 98.7% of all potential DBD donors and 87.4 % of DCD donors were referred to NHS Blood and Transplant for consideration of organ donation. In summary, very few transplantable organs were lost or missed: <https://www.odt.nhs.uk/statistics-and-reports/potential-donor-audit-report/>

VERSION 2 – REVIEW

REVIEWER	Khaled Tafran University of Malaya
REVIEW RETURNED	08-Jun-2019
GENERAL COMMENTS	The authors have addressed the previous comments and the article is publishable now
REVIEWER	Wai Lim Sir Charles Gairdner Hospital, Perth, Western Australia, Australia
REVIEW RETURNED	11-Jun-2019
GENERAL COMMENTS	The authors have adequately addressed my concerns